# Localization in Two-Dimensional Quasicrystalline Lattices

**DOI:** 10.3390/e24111628

**Published:** 2022-11-10

**Authors:** Luis Antonio González-García, Héctor Alva-Sánchez, Rosario Paredes

**Affiliations:** Instituto de Física, Universidad Nacional Autónoma de México, Apartado Postal 20-364, México D. F. 01000, Mexico

**Keywords:** Bose–Einstein condensates, localization in quasicrystals, Gross–Pitaevskii equation

## Abstract

We investigate the emergence of localization in a weakly interacting Bose gas confined in quasicrystalline lattices with three different rotational symmetries: five, eight, and twelve. The analysis, performed at a mean field level and from which localization is detected, relies on the study of two observables: the inverse participation ratio (IPR) and the Shannon entropy in the coordinate space. Those physical quantities were determined from a robust statistical study for the stationary density profiles of the interacting condensate. Localization was identified for each lattice type as a function of the potential depth. Our analysis revealed a range of the potential depths for which the condensate density becomes localized, from partially at random lattice sites to fully in a single site. We found that localization in the case of five-fold rotational symmetry appears for $\left(6E_{R},9E_{R}\right)$, while it occurs in the interval $\left(12E_{R},15E_{R}\right)$ for octagonal and dodecagonal symmetries.

## 1. Introduction

Perfect long-range order with an absence of translational periodicity in unconventional structures in two dimensions, namely the known quasicrystals existent in matter [1,2,3], may be the origin of the well-defined and different physical properties of such materials, compared to those exhibited by crystalline structures. For instance, large variations in surface-related properties [4,5,6] with respect to common metals are found in quasicrystals, including surface energy, mechanical properties such as hardness and coefficient of friction, and thermal and electrical conductivities. This is why, since their discovery in 1984 in aluminum-based alloys, these peculiar solids with quasicrystalline structure represent an alternative for designing materials with specific properties. Up to now, our understanding of the transport properties, such as thermal and electrical conductivity, of quasicrystals has relied mostly on research of a phenomenological character [7,8]. As a matter of fact, quasicrystals have thermal and electrical conductivities that closely resemble those of an amorphous solid. The lack of a crystalline periodicity but perfect rotational symmetry situates quasicrystals in the middle, where well-established theories describing crystals and amorphous materials do not apply.

The study of the transport properties in real quasicrystals is a complex endeavor in the sense that isolating the pure effects of the lattice structure from those associated with electron–electron interactions is an impracticable task. In contrast, ultracold gases are tunable “apparatuses” that offer the possibility of studying the effects of structure and interactions in a separate way [9]. Instead of electrons traveling across the ion cores, large conglomerates of neutral atoms in their lowest energy state move in stationary structures of light, the so-called optical lattices, that can be experimentally set to form crystalline, amorphous, or quasicrystalline patterns in one, two, or three dimensions. Atom–atom interactions in these systems can be modified by means of external fields in such a way that the effective interactions can be decreased or increased, thus emulating electron–electron interactions. Thus, ultracold Bose or Fermi gases confined in optical lattices are an ideal platform where the condensed matter physics can be recreated. As a matter of fact, recent literature on ultracold quantum matter has reported the experimental realization of Bose–Einstein condensates confined in fivefold and eightfold quasicrystalline optical lattices [10,11,12,13]. The relevance of such an experimental success is that it might provide the opportunity to test, as previously demonstrated in the case of crystalline optical lattices, the analogous physics of electrons traveling in layered materials. Regarding the theoretical context, ultracold-gas-confined quasicrystals have been demonstrated to show several phases, including superfluid [14,15], density wave, localized, and supersolid phases [10]. It is important to mention that in the case of the supersolid phases, dipolar interactions are required [16,17,18,19].

The purpose of this investigation was to characterize the emergence of localization in a weakly interacting Bose–Einstein condensate confined in different two-dimensional quasicrystalline lattices as a function of the potential depth that defines their structure. The ground state of the Bose gas was determined from the stationary solution of the Gross–Pitaevskii equation for lattices that have a macroscopic size. For the analysis, we considered lattices with a number of sites in the interval ∼$1.5\times 10^{3}<\Omega <2.5\times 10^{4}$. Similarly to the analysis of the Aubry–André model [20,21], localization in these lattices was tracked in a robust way by means of a statistical analysis using multiple realizations. The index participation ratio (IPR) and the Shannon entropy were the physical quantities used to characterize localization properties. Because of the lattice irregularities, namely the heterogeneous size of lattice sites, we adopted Voronoi diagrams to identify the sites and thus record the variations of the superfluid density across the quasicrystalline structures. The results found in this investigation are in good agreement with the recent observation for an eightfold quasicrystal, in which extended states are found for shallow lattices, while localized states emerge from a certain lattice depth [11].

This paper is organized as follows. In Section 2, we describe the lattices considered in our study, as well as the theoretical approach employed to analyze the influence of the potential depth in localizing the superfluid density. In Section 3, we concentrate on characterizing the transition to localization in terms of two observables: the index participation ratio (IPR) and the Shannon entropy in the coordinate space. These two observables were studied as a function of the potential depth $E_{0}$. We emphasize here that, due to the intrinsic irregularity of the quasicrystalline structures analyzed, we employ the Voronoi construction to determine the inverse participation ratio across the sites. Finally, we present our conclusions in Section 5.

## 2. Model

To investigate the influence of the quasicrystalline symmetry in producing localized states in a weakly interacting Bose–Einstein condensate, we work within the mean field scheme considering the time-dependent Gross–Pitaevskii equation (GPE) that describes the superfluid component ψ(r,t). Through a robust set of numerical calculations for the stationary states of the wave function amplitude ψ(r,t), we shall identify the appearance of localization by evaluating certain properties. The time-dependent equation satisfied by the superfluid component ψ(r,t) is as follows:(1)$$i\hbar \frac{\partial \psi (r,t)}{\partial t}=-\frac{\hbar^{2}}{2m}\nabla_{⊥}^{2}\psi \left(r,t\right)+\left(V_{\rho }^{\epsilon }\left(r\right)+g_{2D}{\left|\psi \left(r,t\right)\right|}^{2}\right)\psi \left(r,t\right),$$
where $\nabla_{⊥}^{2}=\frac{\partial^{2}}{\partial x^{2}}+\frac{\partial^{2}}{\partial y^{2}}$, r=(x,y), $V_{\rho }^{\epsilon }\left(r\right)$ is the potential associated with a particular quasicrystalline structure, and $g_{2D}=\frac{g_{3D}}{\sqrt{3\pi }l_{z}}$ is the effective coupling interaction [22] in 2D [23,24,25,26,27,28], with $g_{3D}=\frac{4\pi \hbar^{2}a_{s}}{m}$, and $a_{s}$ the *s*-wave scattering length in 3D. The label ρ in $V_{\rho }^{\epsilon }\left(r\right)$ serves to identify the quasicrystalline rotation symmetry. The lattices analyzed here have ρ=5,8, and 12 for pentagonal, octagonal, and dodecagonal rotational symmetries, respectively. In the following lines, we point out the particular expressions of the potential $V_{\rho }^{\epsilon }\left(r\right)$ associated with each lattice. The superscript ϵ in the potential amplitude $V_{\rho }^{\epsilon }\left(r\right)$ is a label to identify different realizations. Below we shall provide a detailed description of the meaning of these realizations.

The lattice structure with a pentagonal symmetry can be generated through the following potential [29]:(2)$$V_{\rho =5}^{\epsilon }\left(r\right)=V_{0}^{\epsilon }{\left|\sum_{j=0}^{4}e^{i2G_{j}\cdot r}\right|}^{2},$$
where the vectors $G_{j}$ are given by $G_{j}=\left[\cos\left(\frac{2\pi }{5}j\right),\sin\left(\frac{2\pi }{5}j\right)\right],$ and the value of the potential was chosen to start at zero.

For a quasicrystalline lattice with eightfold symmetry, we have the following [10,11]:(3)$$V_{\rho =8}^{\epsilon }\left(r\right)=V_{0}^{\epsilon }\sum_{j=1}^{4}\cos^{2}\left(3\frac{G_{j}}{2}\cdot r\right),$$
with $G_{1}=\left(1,0\right)$, $G_{2}=\left(0,1\right)$, $G_{3}=\frac{1}{\sqrt{2}}\left(1,1\right)$, and $G_{4}=\frac{1}{\sqrt{2}}\left(-1,1\right)$.

Finally, the formula associated with the dodecagonal quasicrystalline symmetry is as follows:(4)$$V_{\rho =12}^{\epsilon }\left(r\right)=V_{0}^{\epsilon }\left(\cos\left(2\pi x\right)+\cos\left(\sqrt{3}\pi y\right)\cos\left(\pi x\right)+\cos\left(\pi y\right)\cos\left(\sqrt{3}\pi x\right)+\cos\left(2\pi y\right)\right).$$It should be mentioned here that several expressions can be used to describe a dodecagonal quasicrystal, such as those that appear in [30,31]. Quasicrystalline structures are characterized by both the absence of long-range translational symmetry and minima that depend on the site position. These two features suggest that the potential amplitude in $V_{\rho }^{\epsilon }\left(r\right)$, namely $V_{0}^{\epsilon }$, is precisely the important quantity to be varied in order to observe localization. As a matter of fact, the emergence of localization will be identified in terms of the potential depth $E_{0}$ that results from the sum of the contributions in each formula of the quasicrystalline lattices given above.

As is well known, localization in the model introduced by Anderson [32] arises from disorder in form of a random energy shift at each lattice site. While in 1D and 2D, arbitrarily low values of disorder produce localization in all states, in 3D, a threshold of disorder is required to generate localization. The Aubry–André model, which also predicts localization for quasicrystalline lattices in 1D from a given value of the amplitude, is characterized by an incommensurate factor that alters the otherwise perfect crystalline order. Different realizations of quasiperiodicity refer to the election of a phase ϕ∈[0,2π) [20,21]. In this work, to detect localization in a robust way, we adopted a method that, on one hand, allowed us to introduce the element of randomness in the form of a small perturbation of the quasicrystalline structure, and on the other, provided the possibility of taking the average, over a number of realizations, of the physical quantities used to identify localization. Each element of the ensemble was introduced by means of a function ϵ(x,y), which adds/subtracts a small random variation to the potential amplitude through a function ϵ(x,y) (this is why the symbol ϵ in the above formulas refers to different realizations given a fixed value of the potential amplitude $V_{0}^{\epsilon }$). The function ϵ(x,y), defined over the 2D space where the quasicrystalline lattice lies, generates a non-correlated distribution of random numbers varying in the interval [−0.1,0.1] at each point (x,y) in the grid. Thus, the potential amplitude $V_{0}^{\epsilon }$ transforms into $V_{0}^{\epsilon }=V_{0}\left(1+\epsilon (x,y)\right)$. It should be noted here that this small perturbation, which changes the potential amplitude by at most 10 percent with respect to its original value at point (x,y), does not interfere with the well-defined quasicrystalline symmetry of each lattice.

### Energy Scale and Length Units

To perform our numerical simulations, we considered as a reference the standard parameters of an ultracold gas of ${}^{87}$Rb atoms confined in an optical lattice in 2D. The potential amplitude $V_{0}$ was measured in terms of the recoil energy $E_{R}=h^{2}/\left(8ma^{2}\right)$, with a=532 nm as the lattice constant in a square lattice. Note that although in the case of quasicrystalline structures the lattice constant is not defined, *a* represents an experimental reference to properly set the recoil energy. Typical potential depths are in the interval $E_{0}∼\left[4E_{R},16E_{R}\right]$, with $E_{0}$ being the sum of the contributions in each expression for the quasicrystalline lattices (Equations Equation 2–Equation 4). As we argue below, the largest potential amplitude used in our numerical experiments, as well as the values of the effective interaction strength considered ($U_{2D}\propto g_{2D}∼0.01E_{R},\;∼0.10E_{R},\;∼1.1E_{R}$) [19], are such that it guarantees that the mean field Gross–Pitaevskii description is reliable and away from insulating states in the equivalent lattice (tight-binding) model [33]. In other words, there is no competition with Mott states. Moreover, we neglected quantum depletion effects as we are in the T=0 regime. Therefore, as we are deep in the superfluid regime in 2D, the GP theory is an appropriate description to characterize localization of the superfluid component in the quasicrystalline structures.

For the analysis, we considered an ensemble of realizations for the functions ϵ(x,y), for a given lattice composed of Ω sites. Ω represents the number of local minima and absolute minima of the potentials defined by Equations (Equation 2)–(Equation 4). We should mention here that our investigation contemplates a finite size analysis, considering lattices with size Ω, similar to those prepared in current experiments with ultracold atomic gases confined in optical lattices [10]. In this work, we considered $10^{3}<\Omega <2.5\times 10^{4}$.

To describe the ground state of the system, we numerically solved Equation Equation 1 using imaginary time evolution (t⟶iτ), which is equivalent to finding the lowest energy solution or steady-state $\psi \to \psi_{0}$, using the split-step Crank–Nicholson method. This solution represents the amplitude of the macroscopic wave function of the system at T=0 in the lowest energy state. We analyzed the density profiles $\rho \left(x,y\right)=|\psi \left(x,y\right){|}^{2}$, for each quasicrystalline potential. Our approach is limited in the sense that it does not account for depletion of the condensate or non-Poissonian density distributions, and it is a zero temperature model. However, it allows us to study the influence of the quasicrystalline geometry in producing localized states, which is the main purpose of this work. The identification of localization is made from the stationary state by first quantifying the density across the lattice sites as a function of the potential depth $E_{0}$ for each lattice geometry. Then, as described above, to further comprehend the emergence of localization, we investigated the behavior of two properties that are customarily used to identify localization.

## 3. Localization in Quasicrystalline Lattices

In this section, we characterize the transition to localization for the quasicrystalline lattices described in Equations (Equation 2)–(Equation 4). For this purpose, we studied the stationary states of the GPE as a function of the potential depth $E_{0}$, and investigated two observables: the IPR and the Shannon entropy [34]. As is well known, these observables provide signatures of localization in lattices in arbitrary dimensions. In recent literature, the inverse participation ratio of the single particle in momentum space has been calculated to study the localization transition in the ground state of weakly interacting bosons [11]. As written above, our study focuses on a statistical analysis for the observables, in which a number of realizations for given values of the amplitude $V_{0}^{\epsilon }$ and size Ω of the lattice were considered. At this point, it is important to mention that, for lattices with a large number of sites, it was not necessary to have a large number of realizations since fluctuations with respect to the average of a given physical quantity decrease as the system size increases. For our analysis, we computed sets of ∼40 realizations for a given potential amplitude, and for each quasicrystalline structure for lattices with a size $10^{3}<\Omega <6\times 10^{3}$. For lattices with a size $\Omega ∼2.5\times 10^{4}$, we considered just five realizations of ϵ(x,y). The physical quantity collected from raw data, namely the superfluid density at each point, allowed us to select the non-negligible contributions to perform the statistical analysis. To identify those non-negligible contributions, we proceeded as follows. For a given amplitude $V_{0}^{\epsilon }$, only sites with densities larger than 5 percent of the highest density per site were considered for the statistical analysis. Smaller amplitudes were not considered since they are not associated with sites of the quasicrystalline lattice.

It is important to stress here that in order to avoid misinterpretation caused by large potential amplitudes $V_{0}^{\epsilon }$ and wrong use of the GPE approach (that is, values of $V_{0}^{\epsilon }$ resulting in total depths $E_{0}$ for which mean field treatment is not appropriate), we evaluated the correlation function $g_{1}=\frac{1}{z^{2}}\sum_{m\in n.n.}{\left|\int dxdy\;\psi^{*}\left(x,y\right)\psi \left(x+x_{m},y+y_{m}\right)\right|}^{2}$ [35,36], as a function of the potential amplitude, where *z* in this function is the coordination number and *m* is a label that identifies the nearest neighboring sites with respect to a site located in (x,y) position. In addition, $g_{1}$ is a measure of the correlation function between nearest neighbors, and it allows us to establish the range where the potential amplitude can be varied before the system enters into the phase in which the density of the condensate becomes discontinuous across the lattice. For values of the potential depth in the intervals $3E_{R}<E_{0}<10E_{R}$ for the pentagonal quasicrystal, and $3E_{R}<E_{0}<16E_{R}$ for octagonal and dodecagonal geometries, the correlation function remained above zero, thus supporting the GP approach.

The information obtained directly from each realization can be summarized as follows. (1) At potential depths of $E_{0}=3E_{R}$, the condensate density fills the entire space, with the superfluid density distributed across the whole lattice sites in the form of Gaussian peaks. Certainly, the amount of superfluid density depends on the potential depth at each lattice site. At the edges, the density becomes zero as a consequence of the boundary condition (end of the lattice). (2) As the amplitude of the potential increases, Gaussian peaks are not distributed uniformly in the whole lattice; instead, several sites in the lattice show a diminished density, exhibiting peaks with lower amplitudes. As stated above, we fixed an arbitrary criterion to account for the non-negligible peaks contributing to the superfluid density (for a given disorder strength, only peaks with amplitudes larger than 5 percent of the highest amplitude were considered for the statistics). (3) As expected, different realizations labeled by ϵ associated to a given value of the potential amplitude resulted in different distributions of peaks across the lattice. It is worth noting that different realizations have, on average, the same distribution of the superfluid density, but concentrated in different lattice sites. (4) As the potential depth increases, the spatially distributed random peaks become sparse; therefore, in accordance with the condition of constant density, the heights of the peaks become taller. (5) We found a certain range of potential depth for which the scenario of superfluid density distributed across the lattice is replaced by a fragmented density.

### Inverse Participation Ratio

One of the quantities that characterize the structure of an extended state is the participation ratio [37] $p={\left(L^{d}\int {\left|\psi \left(r\right)\right|}^{4}dr\right)}^{-1}$, where *L* is the length of the system in *d* dimensions, and $p∼{\left(l/L\right)}^{d}$ is equal to the fraction of the volume occupied by the wave function with the linear dimension of *l*. However, the inverse participation ratio (IPR) is the quantity most widely used to measure the spatial extent of a localized state, which is given by the absolute value of the fourth power of the wave function [38], $\mathrm{IPR}=\int_{S}{\left|\psi \left(r\right)\right|}^{4}dr$, where the integration region *S* refers to the whole space where the wave function is defined. Originally, the idea behind the definition of the IPR for discrete lattices is that in the opposite extreme situations of extended and localized states, the IPR reaches the values of zero and one, respectively. For macroscopic lattices, an extended state has almost negligible contributions to the probability at each site across, while a localized state has a probability equal to one in a given site and zero in the rest of the lattice. Since our investigation deals with lattices defined in a continuos space, the superfluid fraction also lies in the continuum. As we describe in the next paragraph, a Voronoi diagram is used to properly consider the inherent lattice irregularities of quasicrystalline structures.

To identify the occurrence of localization in the weakly interacting Bose condensate confined in the quasicrystalline lattices, we must consider two factors: first, the non-discrete aspect of the GPE, and second, the fact that the potential that defines the quasicrystalline structures is composed of minima with different depths, and therefore identification of sites requires one to assess in an appropriate way the potential. To properly detect the emergence of localization as a function of the potential depth $E_{0}$, we shall adopt the method of Voronoi diagrams. As we describe below, this method allows us to perform the integration of the fourth power of the wave function in well-defined regions or cells, identified from the potential that generates the quasicrystalline structure. Such regions, which serve to define the boundaries of the integral in the IPR, will be described in the next paragraph.

The first step to determine the IPR is the construction of the Voronoi diagram. The Voronoi diagram, defined by of a set of points or sites $\left\{p_{1},p_{2},\ldots ,p_{n}\right\}$, divides the plane around each point $p_{i}$ into regions $R(p_{i})$, such that all points inside each region are closer (Euclidean distance) to their defining site $p_{i}$ than to any other site $p_{j\neq i}$ [39]. Each $R(p_{i})$ is a convex polygon and is called a Voronoi cell. The region edges are the locus of points equidistant from two sites, and the vertices are points equidistant from three or more nearest sites. The Voronoi diagram is a polygonal partition or tessellation of the plane with no overlapping regions or gaps. This definition implies that the outermost sites of the Voronoi diagram have unbounded Voronoi regions, which can be limited by an arbitrary boundary (e.g., a rectangle) to avoid having infinite-sized areas. In the case of the quasicrystalline lattices, the Voronoi partitions depend on the order of diffraction peak considered. For each quasicrystalline geometry, the Voronoi diagrams were constructed using the local and absolute minima of the potential V(r) as the Voronoi sites using an in-house code written in MATLAB, taking advantage of the built-in function *voronoin* [40]. The number of sites identified was ∼$10^{3}<\Omega <2.5\times 10^{4}$ for the pentagonal, octagonal, and dodecagonal geometries. Variations of about 5 percent in the total number of Voronoi sites in each lattice, associated with the edge of the lattices, were not considered in the calculations. It is important to mention that the Voronoi method was validated using a square lattice—that is, we evaluated the IPR in the square lattice both, through the formula with the integral over the whole surface *S*, and using the Voronoi method.

Figure 1 shows a fragment of the potential associated with the pentagonal, octagonal, and dodecagonal symmetries, as well as its corresponding Voronoi diagrams. The figures on the top show the magnitude of the lattice potential in a density color scheme, with blue and yellow colors corresponding to the maxima and minima, respectively. The red dots indicate the positions of the minima, which served as the sites to define the Voronoi cells. These regions appear in the bottom of Figure 1. It must be noticed that these fragments were chosen from lattices with a large number of sites, and correspond to a region centered in the origin, thus revealing the rotational symmetry of the quasicrystalline structures. We must point out that, although the number of sites of each lattice varies in Figure 1, approximately 240, 190, and 550 sites for the pentagonal, octagonal, and dodecagonal, respectively, are shown at the same size. Interestingly, in the Voronoi diagrams associated with the pentagonal and octagonal quasicrystals, there are some Voronoi cells with red dots apparently being bisected by Voronoi boundaries. In reality, this is an effect caused by two neighboring sites very close to each other, with their corresponding frontier between them.

In our analysis, the IPR was numerically determined for each lattice, as a function of the potential depth $E_{0}$, from the density profile. The boundaries of the integral considered in equation that defines the IPR were those of the Voronoi cells (see fragments of the Voronoi lattice shown in Figure 2). In Figure 2, we illustrate in a density color scheme the whole space of the Voronoi regions and the magnitude of the IPR across the cells, associated with a single realization of ϵ(x,y). The top, middle, and bottom figures correspond to pentagonal, octagonal, and dodecagonal quasicrystals respectively. At the top of each figure, the value of the potential depth associated with each realization is indicated. As one can see from the color bars on the right in each figure, the size of the IPR strongly depends on the potential depth. While shallow potential depths, such as $E_{0}=3E_{R}$, produce Voronoi lattices that have cells with values of the IPR uniformly distributed and very close to zero (∼$\times 10^{-8}$), as the potential depth increases, isolated regions with large values of IPR are found in the Voronoi lattices (∼10${}^{-4}$). These results indicate that the opposite extended and localized states are characterized by having uniform near-zero values of the IPR across the whole Voronoi lattices, and sharply localized superfluid density in a few lattice sites, respectively. The sectors included in Figure 2, which are the same for any given quasicrystalline structure, were intentionally chosen in a different area to those shown in Figure 1 to illustrate that the superfluid density is distributed across the lattice regardless of the observation region. In short, the behavior shown in Figure 2 illustrates the emergence of localization as a function of the potential depth $E_{0}$. It is important to state that these plots correspond to a particular realization ϵ, and as stated above, different realizations of ϵ have, on average, the same number of sites with non-negligible superfluid density, but distributed in different sites.

In order to recognize the emergence of localization as a function of the potential depth, the information of the IPR associated with each realization ϵ→ϵ(x,y) and a given value of the potential depth were organized in the form of histograms, which we constructed as follows. The horizontal axis of each histogram was divided into 40 equally spaced intervals, being the extrema, the minimum, and maximum values of the IPR. Then, we counted the number of Voronoi cells with values of IPR ranging in the identified intervals and used this outcome to construct the histogram. In other words, the bars of each histogram are the number of Voronoi regions with a value of IPR in a given interval. Then, to obtain a representative histogram with all of the statistics (that is, a histogram considering the total number of realizations), we averaged the histograms for a given value of the potential depth $E_{0}$. As mentioned above, 40 realizations were performed for $\Omega \in [10^{3},6\times 10^{3}]$ sites, while 5 realizations were considered for $\Omega ∼2.5\times 10^{4}$. To illustrate the way in which the information of the IPR is arranged in the histograms, in Figure 3, we show in a logarithmic scale the histograms associated with the average over the realizations for $\Omega \approx 6\times 10^{3}$ sites, for different values of the potential depth. As in the case of the previous figure, the rows from top to bottom correspond to pentagonal, octagonal, and dodecagonal quasicrystals. The colors used in the histograms match those of the Voronoi diagrams of Figure 2 to ease visual interpretation.

As can be seen from Figure 3, for shallow potential depths ($3E_{R}$), the histograms exhibit dense distributions, thus implying that most of the sites are occupied and correspondingly that the superfluid is non-localized. Then, as the potential depth is increased ($7E_{R}$ for the pentagonal quasicrystal and $12E_{R}$ for the octagonal and dodecagonal geometries), the histograms acquire a different aspect. For large values of potential depth (see figures on the right column), just a few of the sites are occupied, and therefore the histograms are composed of few bars. Note that in the horizontal axis, the values of the IPR change by many orders of magnitude as the potential depth grows, while figures in the left column have IPR values ∼$10^{-8}$, figures in the central column carry on IPR values ∼$10^{-7}$, and finally, for the histograms on the left column, the values of the IPR are ∼$10^{-4}$.

This behavior indicates that the transition to localization starts at values of $E_{R}$ around those chosen in the center column. Indeed, we observe from Figure 3 that the histograms associated with the IPR contributions of Figure 2 show a sparse distribution as the potential amplitude is increased. To discern localization, we must track how dense histogram distributions become sparse. From the statistical moment distribution of the IPR, one can observe that the kurtosis is the moment that better captures the transition. Thus, we use this quantity to quantify this transition, since it measures the "tailedness" of a distribution, or how outlier-prone a distribution is. In Figure 4, we plot the kurtosis for the pentagonal, octagonal, and dodecagonal lattices as a function of the amplitude $V_{0}^{\epsilon }$. Different symbols in each plot correspond to different numbers of sites Ω, with diamonds, squares, triangles, and circles labeling $\Omega =2\times 10^{3},4\times 10^{3},6\times 10^{3}$, and $2.5\times 10^{4}$, respectively. As one can see from these figures, there is a starting value of the potential amplitude for which the kurtosis becomes different from zero, followed by a region in which the kurtosis increases until a maximum value is reached. As expected, this maximum equals to the limit value of the kurtosis, which signals that the dense histogram distribution is turned into a single bar specifically when the superfluid is located in a single site. One can identify a region of the potential amplitudes for which localization is present. For the pentagonal, octagonal, and dodecagonal lattices, the intervals in units of $E_{R}$ are [6,10], [12,16], and [12,16], respectively. From the behavior of the kurtosis, one can identify a crossover region of the potential amplitude in which the smooth transition from extended to localized states occurs. From Figure 4, we observe that for the largest value of Ω considered (that is, $\Omega =2.5\times 10^{4}$), the smooth behavior observed in the kurtosis for Ω=1.5, 4, and 6 $\times 10^{3}$ is replaced by an almost abrupt change starting from certain values of potential depth (see, for instance, $E_{0}=7E_{R}$ in the case of pentagonal quasicrystal and $E_{0}=13E_{R}$ for octagonal and dodecagonal symmetries). Numerical calculations for larger values of lattice sites ($\Omega ∼10^{5}$) are impracticable to compute.

We should stress that the existence of localization in the weakly interacting condensate confined in the two-dimensional lattices does not arise from the random fluctuations enclosed in ϵ, but from the quasicrystalline structures as the potential amplitude surpasses certain depth.

## 4. Shannon Entropy of BEC Matter Waves

Another suitable physical property to study the localization phenomena in a theoretical context is the Shannon entropy [34]. This observable has been used to investigate the localization transition of matter waves in Bose–Einstein condensates within the framework of the Gross–Pitaevskii equation. Among other systems, quasiperiodic lattice potentials in one dimension [41,42] and dipolar condensates in three dimensions in the presence of anisotropic harmonic confinement [43] have corroborated the influence of the interplay of kinetic energy, interatomic interactions, and disordered potentials. For lattice models, the analysis inside the tight binding approximation [42,44] includes both periodic and antiperiodic boundary conditions.

The purpose of this section is to characterize the localization of Bose–Einstein condensates lying in quasicrystalline structures by tracking the Shannon entropy. Here, we should point out that other thermodynamic properties have been investigated in the past [45]. From the information provided by the stationary density profiles as a function of the potential depth for each quasicrystalline lattice, we present here the analysis of the Shannon entropy for the coordinate space. For a continuous probability distribution $\rho \left(x,y\right)=|\psi \left(x,y\right){|}^{2}$, the Shannon information entropy in coordinate space is defined by [42,43,46,47] as $S_{\rho }=-\int \rho \left(x,y\right)\ln\rho \left(x,y\right)dxdy$, with ∫ρ(x,y)dxdy=1. We notice from this last equation that $S_{\rho }$ is an extensive quantity, which depends on the lattice size. Now, let $\psi (k_{x},k_{y})$ be the Fourier transform of ψ(x,y); then, the Shannon entropy associated with $\gamma \left(k_{x},k_{y}\right)=|\psi \left(k_{x},k_{y}\right){|}^{2}$ in the momenta space is given by $S_{\gamma }=-\int \gamma \left(k_{x},k_{y}\right)\left(\ln\gamma (k_{x},k_{y})\right)dk_{x}dk_{y}$, with $\gamma \left(k_{x},k_{y}\right)={\left|\psi \left(k_{x},k_{y}\right)\right|}^{2}$, and $\int \gamma \left(k_{x},k_{y}\right)dk_{x}dk_{y}=1$.

The physical meaning of $S_{\rho }$ and $S_{\gamma }$ is as follows. The Shannon entropy in the space of positions corresponds to the measurement of the uncertainty of the location of the particles in space, in such a way that the lower this entropy is, the more concentrated the wave function becomes. In other words, when the position uncertainty is smaller, the accuracy in predicting the localization of the system grows. On the other hand, the meaning of the Shannon entropy in the space of momenta corresponds to measuring the uncertainty of predicting the momentum of the particles, and this entropy provides information on propagation (high entropy values) and regeneration (low entropy values) of well-localized wave packets [46]. An important fact that must be taken into account is that if one entropy value decreases, the other increases, and vice versa. Since $S_{\rho }$ and $S_{\gamma }$ are complementary quantities, we concentrate here on analysis of the former as a function of the potential depth.

In Figure 5, we show the results obtained for Shannon entropy in coordinate space for the analyzed lattices. Different curves in each panel correspond to several values of Ω. This size was determined, as in the previous section, by means of the Voronoi diagrams associated with the minima of the potential that define a given structure. As expected, each panel in Figure 5 reveals the extensive character of $S_{\rho }$, since the maximum value of the entropy is proportional to the size of the lattice. In addition, we observe that the behavior of $S_{\rho }$ shows the characteristic behavior that allows one to detect localization as a function of the potential depth. Note that $S_{\rho }$ is approximately constant for a certain interval of the potential depth, and then it starts to decline in a continuous way with increasing potential depth. As can be appreciated from Figure 5, depending on the particular quasicrystalline geometry, $S_{\rho }$ starts to diminish at different values of $E_{0}$, but qualitatively, its behavior in each case is the same. Looking back at Figure 4, the values of the potential depth for which kurtosis registers the change correspond to those of Figure 5, in which $S_{\rho }$ also exhibits a transition. Therefore, the behavior of the IPR kurtosis and the entropy reflect the occurrence of the transition to localization in quasicrystalline lattices as a function of the potential depth.

Besides the analysis for different values of the lattice size, we also performed an investigation of the Shannon entropy as a function of the interaction strength. As stated above, the values of the effective interaction were chosen away from insulating states in the equivalent lattice [48]. In Figure 6, we plot the Shannon entropy for $\Omega =6\times 10^{3}$ and three different values of the interaction amplitude. As can be seen from this figure, the Shannon entropy is qualitatively similar, disregarding the size of the interaction. The most evident quantitative effects arise in the case of the dodecagonal lattice.

Section 3 and Section 4 allow us to identify how localization in the analyzed quasicrystalline lattices occurs as the potential depth exceed a certain size. These results are in accord with a similar study conducted recently by M. Sbroscia and collaborators [11] for an eightfold quasicrystal.

## 5. Conclusions

In this work, we have studied the localization phenomenon of a weakly interacting Bose–Einstein condensate lying in optical lattices with quasicrystalline structure. We considered lattices with pentagonal, octagonal, and dodecagonal rotation symmetries, with a number of sites in the interval $1.5\times 10^{3}<\Omega <2.5\times 10^{4}$. For the analysis, we quantified the behavior of two observables, the index participation ratio (IPR), and the Shannon entropy in coordinate space $S_{\rho }$ as a function of the potential depth $E_{0}$. These observables were investigated from a statistical analysis considering an ensemble of realizations for a fixed amplitude $E_{0}$. To properly quantify the superfluid density across the lattice sites, besides considering an ensemble of realizations, a Voronoi diagram was constructed to track the variation of the superfluid density site by site. The observables analyzed confirmed a smooth crossover from extended to localized states as a function of the lattice depth. While localization in the case of fivefold rotational symmetry appears for $\left(6E_{R},9E_{R}\right)$, it occurs in the interval $\left(12E_{R},15E_{R}\right)$ for octagonal and dodecagonal symmetries. The analysis for different values of the effective mean field interaction, performed for $\Omega =6\times 10^{3}$ sites, showed the same qualitative behavior for the Shannon entropy. In fact, the quasicrystalline structure dominates over the effective mean field interactions to generate localization as the potential depth is increased.

Our results, relevant within the context of condensed matter, could also be tested in samples of degenerate gases confined in optical lattices with a quasicrystalline structure as revealed in [11] for an eightfold symmetry. In this case, an important aspect to consider is that the potential depth created from the interfering lasers arrays must be such that the fraction of the superfluid density exists in the quasicrystalline structures constituted of minima with different amplitudes. In our numerical calculations, we used values of the potential depth for which a Mott phase does not compete with the emergence of localization [33]. Even more, to ensure that the superfluid is far from the interaction-driven insulating states, effective values of the interaction strength must not exceed certain values of the potential depth (∼$10E_{R}$ for pentagonal quasicrystal symmetry, and $16E_{R}$ for octagonal and dodecagonal geometries). An additional fact that must be pointed out is that the two-dimensional quasicrystals here analyzed are the counterpart of the study of Anderson localization in lattices with random disorder. In both cases, localization emerges as the “disorder amplitude” grows. However, the main difference between our results and those within the Anderson picture is that localization appears from a certain potential depth of the quasicrystalline structures, as in the case of the Aubry–André model, and not for an arbitrarily small disorder strength, as in the case of random disorder. Density wave and supersolid phases are beyond the scope of the present work since long-range interactions are absent in our approach.

## Figures and Tables

**Figure 1 entropy-24-01628-f001:**
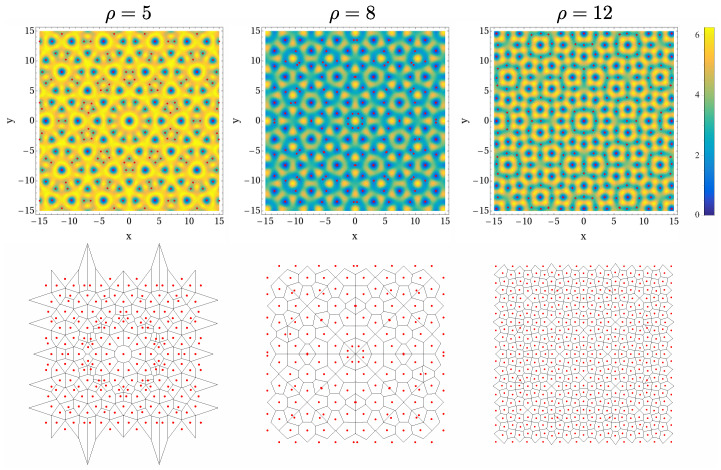
(Color online.) Fragments of the lattice potential for the pentagonal (**left**), octagonal (**center**), and dodecagonal (**right**) quasicrystalline structures. Top figures show the potential maxima and minima in a density color scheme, with the red dots indicating the minima. Bottom figures are the associated Voronoi diagrams of the top figures. $\Omega ∼10^{3}$. Parameters are: $E_{0}=6E_{R},U=0.01E_{R}$. It is important to point out that the fragments chosen correspond to the center of the lattices.

**Figure 2 entropy-24-01628-f002:**
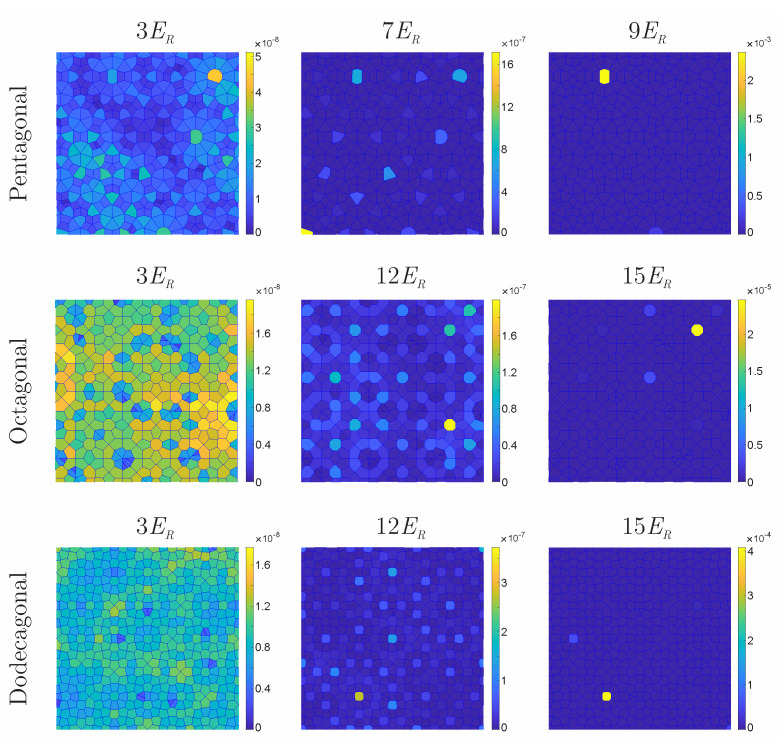
(Color online.) Fragments of the Voronoi regions for pentagonal (**top**), octagonal (**center**), and dodecagonal (**bottom**) quasicrystalline structures. The color scheme in the bars on the right of each figure indicates the value of the IPR at each Voronoi cell, for the potential depth indicated at the top border of each figure.

**Figure 3 entropy-24-01628-f003:**
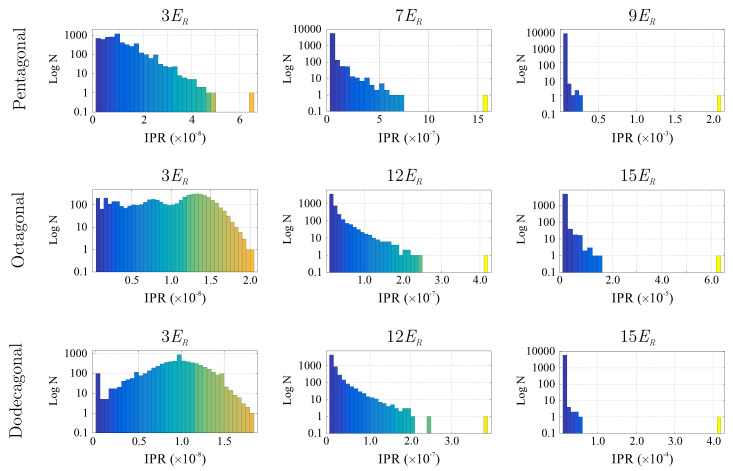
(Color online.) Histograms of the IPR values for pentagonal (**top** row), octagonal (**center** row), and dodecagonal (**bottom** row) quasicrystalline lattices. At the top of each figure appears the value of the potential depth in units of recoil energy $E_{R}$. The values of the IPR correspond to 40 realizations for lattices with $6\times 10^{3}$ sites.

**Figure 4 entropy-24-01628-f004:**
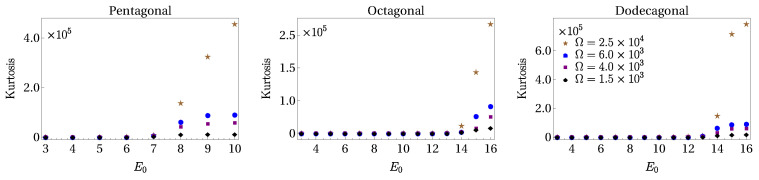
(Color online.) Kurtosis as a function of the potential depth for pentagonal (**left**), octagonal (**center**), and dodecagonal (**right**) quasicrystalline structures. Symbols in each panel identify different values of the site lattice number Ω, which is common to all panels. The effective coupling interaction is $U=0.01E_{R}$.

**Figure 5 entropy-24-01628-f005:**
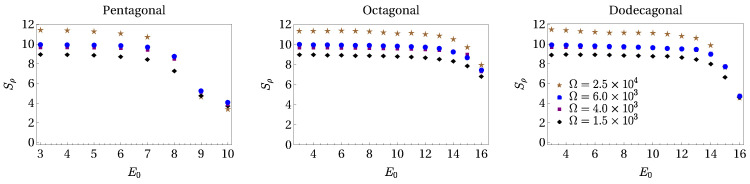
(Color online.) From **left** to **right**, we plot the Shannon entropy in coordinate space $S_{\rho }$ as a function of potential depth for pentagonal, octagonal, and dodecagonal quasicrystalline lattices. Symbols in each panel identify different values of the site lattice number Ω, which is common to all panels. The effective coupling interaction is $U=0.01E_{R}$.

**Figure 6 entropy-24-01628-f006:**
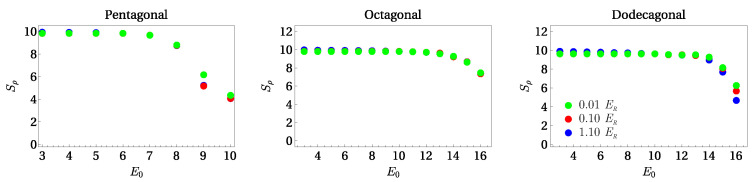
(Color online.) From **left** to **right**, we plot the Shannon entropy in coordinate space $S_{\rho }$ as a function of potential depth for pentagonal, octagonal, and dodecagonal quasicrystalline lattices. Symbols in each panel identify different values of the effective mean field interaction. The symbols correspond to $U=0.01E_{R},\;0.1E_{R}$, and $1.1E_{R}$, respectively.

## Data Availability

Not applicable.

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
