# Peer review of "Localization in Two-Dimensional Quasicrystalline Lattices"

_entropy, 2022, doi:10.3390/e24111628_

Round 1
Reviewer 1 Report
The manuscript discuss the localization problem for weakly interacting Bose gas in two dimensional quasicrystal lattices. The IPR and the Shannon entropy are used to characterize state localization properties. The work is interesting. However, several things should be further considered :
1. The paper is not well written. It should be concise and comprehensive. For example, the title can change to `` Localization in two dimensional quasicrystalline lattices’’. Very long sentences may change to short ones. In the third paragraph of Introduction, the detail of calculation can be described in the next section. The Conclusion section is cumbersome.
2. Localization due to disorder is the main result. Traditionally, it should be pointed out where the transition happens.
3. In Abstract, the authors say that `` Among the three geometries studied, the five-fold rotational symmetry demonstrated to become a structure that easily localizes’’ . In text, the reason should be given.
4. The potential depth $E_0$ is an important parameter. The expression is necessary.
5. The Voronoi diagram method is used to divide continuous space. The references using the method to characterize state localization properties should be cited if they exist. In fact, in Eqs.(5) and (6), the expressions of IPR and $S_\rho$ are defined in continuous space. For them, the continuous space is not necessary to divide into discrete ones.
6. Lines 117-118: ``...a phase $\phi\\in[0,2\pi)$ that identifies a given realization of quasiperiodicity [30,31].’’ Just from the sentence, one realization of potentials is not easily gotten.
7. Lines183-185: ``For a given amplitude $V_0^\epsilon$ , only sites having densities larger than 5 percent of the highest density per site were considered for the statistical analysis.’’ What will happen if all data are chosen.
8. Lines195-197: ``For values of the potential depth in the intervals $3E_R<E_0<10 E_R$ for the pentagonal quasicrystal, and $3E_R<E_0<16E_R$ for octagonal and dodecagonal geometries, the correlation function remained above zero, thus supporting the GP approach.’’ Why not consider the condition that $E_0<3E_R$ ? What is the reason that ``the correlation function remained above zero, thus supporting the GP approach’’.
9.Ref. [43] is just the reference [39].
Author Response
In the attached file we include the response to the Referee report.

Reviewer 2 Report
Please see the attached file

Author Response
In the attached file we included the response to the Referee report.

Round 2
Reviewer 1 Report
The author answer all comments.